# Taxonomic Diversity and Interannual Variation of Fish in the Lagoon of Meiji Reef (Mischief Reef), South China Sea

**DOI:** 10.3390/biology13090740

**Published:** 2024-09-21

**Authors:** Yuyan Gong, Jun Zhang, Zuozhi Chen, Yancong Cai, Yutao Yang

**Affiliations:** 1College of Marine Living Resource Sciences and Management, Shanghai Ocean University, Shanghai 201306, China; gongyuyan@scsfri.ac.cn; 2South China Sea Fisheries Research Institute, Chinese Academy of Fishery Sciences, Guangzhou 510300, China; zhangjun@scsfri.ac.cn (J.Z.); onion-20062006@163.com (Y.C.); tao@scsfri.ac.cn (Y.Y.); 3Key Laboratory for Sustainable Utilization of Open-Sea Fishery, Ministry of Agriculture and Rural Affairs, Guangzhou 510300, China

**Keywords:** coral reef fish, species composition, inclusion index at taxonomic level, genus–family diversity index, taxonomic distinctness index

## Abstract

**Simple Summary:**

Coral reef fish diversity forms the foundation of their adaptability. A comprehensive understanding of coral reef fish diversity is crucial for the conservation of both coral reefs and their associated fish species. In this study, we employed three taxonomic diversity indices to analyze the characteristics and changes in taxonomic diversity within a representative coral reef (Meiji Reef) located in the South China Sea from 1998 to 2018. Fish species richness played a significant role in shaping the diversity of coral reef fish families and genera within the South China Sea region. Contrary to findings from 1998–1999, there had been a relative increase in fish family diversity at Meiji Reef during 2017–2018. Distribution patterns among different coral reef fish communities across taxonomic levels exhibited varying degrees of unevenness within the South China Sea region. The changes observed in the taxonomic distinctness index between 1998–1999 and 2016–2018 indicated greater dissimilarity among species relationships and reduced taxonomic uniformity.

**Abstract:**

Coral reef fish are important groups of coral reefs, which have great economic and ecological value. Meiji Reef is a representative tropical semi-enclosed atoll in the South China Sea, with rich fish resources. Based on the data from hand-fishing, line-fishing, and gillnet surveys of fish in Meiji Reef from 1998 to 2018, this study summarized the fish species list of Meiji Reef and analyzed the species composition, inclusion index at the taxonomic level (TINCL), genus–family diversity index (G–F index), average taxonomic distinctness index (Δ^+^), and variation in taxonomic distinctness (Λ^+^) and their changes. The results revealed that from 1998 to 2018, there were 166 reef-dwelling fish species on Meiji Reef, belonging to 69 genera, 33 families, and 11 orders, of which 128 species were from 20 families of Perciformes, accounting for 77.10% of the total cataloged species. Regarding the dependence of fish on coral reefs, there were 155 reef-dependent species or resident species (accounting for 93.37%) and 11 reef-independent species or wandering species (accounting for 6.63%). The TINCL of the order, families, and genus of fish in Meiji Reef were very high. The genus diversity index (G index), family diversity index (F index), and G–F index of fish in Meiji Reef were very high, and the G index of fish in Meiji Reef in 1998–1999 was higher than that in 2016–2018. The Δ^+^ and Λ^+^ values of fish in Meiji Reef from 1998 to 2018 were 56.1 and 148.5, respectively. Compared with 1998–1999, Δ^+^ and Λ^+^ of fish increased during 2016–2018, reflecting that the relatives of fish in Meiji Reef became further distant, and the uniformity of taxonomic relationships among species decreased. The research findings indicated that fish exhibited a high taxonomic diversity in Meiji Reef; however, it also revealed significant fluctuations in the fish diversity of Meiji Reef over an extended period, emphasizing the urgent need for timely protection measures. This investigation significantly contributes to our comprehension of the intricate dynamics governing fish species within Meiji Reef and holds broader implications for biodiversity conservation in tropical marine ecosystems.

## 1. Introduction

Coral reefs are internationally recognized for their remarkable biodiversity and productivity, serving as vital breeding grounds for a multitude of marine organisms. They play a pivotal role in upholding global marine biodiversity and sustaining the supply of biological resources [1,2]. Coral reef fish are crucial components of coral reefs, possessing significant economic value and playing a vital ecological role in the material cycling and energy flow within these ecosystems [3,4,5]. The species composition of coral reef fish not only determines the structure of coral reef fish communities but also underpins their multifunctionality. However, over the past few decades, coral reefs have experienced a persistent decline along with diminishing fish stocks, leading to widespread concerns [6]. The decline or loss of diversity within these communities can compromise community stability and impede the ecosystemic services of coral reefs, resulting in species extinction, collapses in food chains, invasions and outbreaks of pests, declines in fish biomass, etc. [7,8,9]. Therefore, measuring and assessing the diversity of coral reef fish is crucial for conserving and promoting the sustainable development of both coral reefs and their associated fish populations [10,11]. Traditional species diversity indices, such as the Margalef species richness index, Shannon–Wiener diversity index, and Pielou evenness index, are of great significance in evaluating the structure of coral reef fish and their changes. These indices serve as important indicators of overall community diversity levels; however, their application is sometimes limited when comparing fish community diversity across different spatial areas or historical data due to inconsistencies in sampling methods, survey duration, or sample sizes [12,13,14].

In the investigation of fish community diversity, analysis solely at the species level is insufficient due to an imbalanced distribution across taxonomic orders, families, genera, and species. Furthermore, there exists taxonomic diversity alongside species richness, abundance, and evenness within a community [15,16,17]. Therefore, investigating community diversity at the taxonomic level serves as a crucial complement to conventional species diversity indices. For instance, taxonomic diversity indices encompass the inclusion index at the taxonomic level (TINCL), genus–family diversity index (G–F index), average taxonomic distinctness index (Δ^+^), and variation in taxonomic distinctness (Λ^+^). The Δ^+^ and Λ^+^ are widely employed as a measure of taxonomic diversity indices [18,19,20]. The notable advantage of taxonomic diversity lies in its exclusive consideration of species occurrence within the study area, rendering it independent of sampling methods and sample size. This results in a lower standardization requirement for sample data, making it more robust when dealing with unknown sampling nature or inconsistent collection methods. Moreover, this approach is particularly conducive to comparative studies on community diversity characteristics across different regions, habitats, or historical data [21,22,23]. For instance, based on fish samples collected in 2018 and 2019, Li et al. reported a higher number of fish species at Qilianyu Island compared to Meiji Reef, along with correspondingly higher G, F, and G–F indices [23]. These findings indicated that the fish diversity at Qilianyu Island surpassed that of Meiji Reef both in terms of family and genus levels.

The coral reef, as the most distinctive ecosystem in the South China Sea, exerts a direct influence on the ecological characteristics of this region and plays a crucial role in upholding marine life diversity not only within the South China Sea but also throughout offshore China [24,25]. Over 2000 species of coral reef fish have been documented, constituting more than two-thirds of the recorded fish diversity in the South China Sea [26,27]. In addition to their crucial role in biodiversity conservation, coral reefs in the South China Sea also provide significant ecological and economic benefits, including fishery support, tourism attraction, and coastal protection services [24,25]. Numerous studies have been conducted on the species composition, species diversity, and taxonomic diversity of coral reef fish communities in the South China Sea, which holds significant implications for elucidating the structure and functionality of these communities within this region [21,28,29,30]. Chen et al.’s results indicated that fish species diversity in coral reef water was significantly different compared with those either in the northern or southwestern continental shelf waters of the South China Sea, and fish species in similar habitats had higher similarity [29]. The findings of Li et al. demonstrate that the distribution patterns of fish diversity in coral reef waters in the South China Sea are characterized by higher species diversity and lower taxonomic diversity, aligning with large-scale trends [21]. However, there is a paucity of studies on the taxonomic diversity of fish in different historical periods. Meiji Reef, located in the South China Sea, represents a typical tropical coral reef with abundant fish resources and has been an early focus area for coral reef fish surveys. In this study, we conducted a comprehensive analysis of the taxonomic composition and diversity of fish species inhabiting Meiji Reef using survey data from 1998 to 2018 in the South China Sea. Our aim was to enhance our understanding of coral reef fish communities while providing valuable tools for assessing their diversity, ultimately promoting effective conservation measures for both coral reefs and associated fish populations.

## 2. Methods and Materials

### 2.1. Study Area and Fish Specimen Collection

The fish specimens were collected in Meiji (9°55′ N, 115°32′ E) in the SCS. The Reef is a typical atoll, with an oval shape and shallow tropical waters (Figure 1). It spans 9 km in length (east–west) and 6 km in width (north–south), covering a total area of 56.6 km^2^. The reef features a large lagoon, reaching a maximum depth of 30 m, which connects to the open ocean through three channels. In the northwest lies a reef flat measuring approximately 3 km long and 0.8 km wide, while the southeast boasts a reef flat that exceeds 4 km in length and spans 0.3 km in width [31]. Meiji Reef and its fish are not currently part of any protected area.

Fish specimens from Meiji Reef were collected by the South China Sea Fisheries Research Institute of the Chinese Academy of Fishery Sciences in 1998–1999, 2012, and 2016–2018, coinciding with a significant marine heatwave event triggered by the El Nino period in both 1998 and 2016 [32]. Three types of fishing gear were used for sampling: hand line, longline, and gillnet (Table 1 provides further details). Different fishing gear was designed for specific sizes and feeding groups of species. Hand-line and longline gear allowed smaller fish to escape, protecting fish larvae. Gillnet gear covered a wider range of feeding habits and sizes, providing a more comprehensive assessment of fish diversity. Initially, hand-line, gillnet, and longline gear was used in early surveys. However, longline gear was discontinued to protect larger fish like sharks while a hand line and gillnet were retained for consistent survey methods. Surveys conducted in 1998 and 1999 took place on board the vessels R/V Fisheries Administration (300 GT, measuring 44.40 m long and 8.00 m wide) and F/V Yueyu 730 (98 GT, measuring 26.50 m long and 5.30 m wide). From 2012 to 2018, surveys were carried out on board the R/V Nanfeng (1537 t GT, measuring 66.66 m long and 12.40 m wide), which was equipped with a motorboat for sampling purposes (1 GT, measuring 7.85 m long and1.50 m wide; powered by a compression-ignition internal combustion engine) [31].

The majority of species were identified in the field, with only those specimens that could not be identified on-site being preserved and subsequently identified in the laboratory. The collection of fish samples was limited to a relatively small number, typically a few hundred at most, and conducted only twice a year, ensuring minimal impact on the fish population. The license file number assigned to our samples is “Ethical approval number: SCSFRI Document number 40/2016”. The coral reef fish samples were collected for the purpose of coral reef fishery resource assessment and scientific management studies in the South China Sea.

The working depth of hand lines with barbed hooks was ~20 m, with fresh shrimp as bait, and sampling was performed during the day (08:00–10:00 and 14:00–18:00). The working depth of gillnets was −20 m, and sampling was usually performed during the day, although occasionally extended to the next morning (06:00). The longline fishing operation targeted the shallow area of the reef edge (10–160 m) and lagoon, utilizing frozen cod or flying fish as bait. Surveys conducted in 1998 and 1999 employed this method, commencing fishing activities in the evening (18:00) and extending until the following morning (06:00). Four sampling sites were established for each survey to cover the lagoon, with GPS used to ensure consistency in site selection [31]. The number of stations was allocated according to the depth, with 1 station at a 5 m depth, 2 stations at a 10 m depth, and 1 station at a 20 m depth. Fish were collected at each site using a hand line and gillnet from 1998 to 2018 [31]. The time, site, and fish catcher of each specimen were recorded. Specimens were immersed in seawater and frozen (−20 °C) for shore-based analysis. Each specimen was identified to the lowest taxonomic category based on morphological characteristics [33,34,35,36,37].

According to the dependence of fish on the coral reef, we categorized coral reef fish into two groups in this study: reef-dependent fish (resident species) and reef-independent fish (wandering species) [38]. Reef-dependent fish primarily inhabit coral reefs, where they form a dominant population. These species are known for their affinity toward reefs and constitute the main component of the coral reef fish community [39]. They often exhibit vibrant colors or intricate patterns, such as parrotfish, grouper, and butterflyfish, among others. On the other hand, reef-independent fish only sporadically occupy coral reef habitats and predominantly reside in other ecosystems outside of coral reefs (e.g., pelagic deep ocean). They represent a small proportion in terms of species diversity or abundance within the coral reef fish communities.

### 2.2. Data Analysis

#### 2.2.1. Inclusion Index at Taxonomic Level

In this study, the inclusion index at the taxonomic level (TINCL) was employed to analyze the concentration of fish species distribution at each taxonomic level and investigate the diversity of fish composition as well as the interrelationship among different taxonomic levels [22]. A higher TINCL value indicates a greater number of taxonomic groups at the species level (genus or family) belonging to a specific taxonomic group at the genus level (family or order), suggesting a more concentrated distribution of fish species and closer relationships. Conversely, a lower TINCL value implies fewer families (genera and species) included in an order (family and genus), indicating a more dispersed distribution of fish species and distant relatives. The temporal variation in the taxonomic composition of reef fish at Meiji Reef was examined by comparing the TINCL values across different time periods. The expression for TINCL is
(1)$${TINCL}_{i}=(1/N_{j})\sum_{j=1}^{N_{i}}C_{kj}(k<i)$$
where N_i_ is the number of taxonomic elements at level i. For instance, if i denotes the order level, and there are 10 orders in the fish composition, then N_i_ equals 10. C_kj_ represents the number of elements at j*th* k-class classification stage. In this study, we analyzed three levels of classification: order, family, and genus represented by 3, 2, and 1, respectively.

#### 2.2.2. Genus–Family Index

The genus–family diversity (G–F diversity) index is a quantitative measure of species diversity in a given region, which takes into account the number of species belonging to different families and genera. This index reflects the diversities at both family and genus levels, providing an efficient and rapid assessment of species diversity that fulfills the requirements for biodiversity measurement. The species diversity index at the family (D_F_) level is calculated as follows [15,23]:(2)$$D_{F}=\sum_{k=1}^{m}D_{FK}$$
(3)$$D_{FK}=-\sum_{i=1}^{n}p_{i}lnp_{i}$$

In the formula, p_i_ = s_ki_/S_k_; S_k_ and s_ki_ are the number of species in family k and the number of species in genus i of family k, respectively; n and m are the number of genera in family k and the number of families in the phylum, respectively; and D_FK_ is the diversity of species in family k.

The species diversity index at the genus (D_G_) level is calculated as follows [15,23]:(4)$$D_{G}=-\sum_{j=1}^{p}q_{i}lnq_{i}$$

In the formula, q_j_ = s_j_/S; S, s_j_, and p are the number of species in the phylum, the number of species in genus *j* in the phylum, and the number of genera in the phylum, respectively.

The standardized G-F index (D_G-F_) is calculated as follows [15,23]:(5)$$D_{G–F}=1-D_{G}/D_{F}$$

If all families in an assemblage are monotypic (namely D_F_ = 0), the G–F index will be 0 (namely D_G–F_ = 0). The more non-monotypic families there are, the higher the G–F index is, ranging 0 ≤ D_G–F_ ≤ 1.

#### 2.2.3. Average Taxonomic Distinctness Index and Variation in Taxonomic Distinctness

The average taxonomic distinctness index (Δ^+^) is defined as the average taxonomic path length between any two randomly chosen species in an assemblage. Within a certain range, the higher the Δ^+^, the farther apart the taxonomic affinities between species in the assemblage, while the opposite is also true [18,20,23]. The variation in taxonomic distinctness (Λ^+^) is the variance of these pairwise path lengths and reflects the unevenness of the taxonomic tree and complements the Δ^+^. A larger Λ^+^ index indicates that the variation in the taxonomic distinctness of species in an assemblage is greater, while the opposite is also true. Δ^+^ and Λ^+^ are calculated as follows:(6)$${∆}^{+}=(\sum \sum_{i<j}\omega_{ij})/[S(S-1)/2]$$
(7)$$\Lambda^{+}=(\sum \sum_{i<j}(\omega_{ij}-{∆}^{+})/[S(S-1)/2]$$
where *w_ij_* is the difference in weighting resulting from the branch length between species *i* and *j*. *S* is the number of species. The branch weight values of the weighted path length among the six taxonomic levels of phyla, class, order, family, genus, and species are 100.000, 83.333, 66.667, 50.000, 33.333, and 16.667, respectively [20,23].

The taxonomic diversity index does not necessitate the uniformity of fishing gear. In this study, the comprehensive fish specimens obtained with a longline, gillnet, and hand line were utilized to assess the TINCL index, G–F index, Δ^+^ index, and Λ^+^ index of the fish community at Meiji Reef.

If the original data were consistent with a normal distribution, we employed a *T*-test to analyze the interannual variation of the TINCL index, G–F index, Δ^+^ index, and Λ^+^ index. Additionally, we utilized the Pearson correlation coefficient to assess the correlation between the diversity index and species richness. In cases where significant deviations occurred, we applied non-parametric tests (Wilcoxon rank sum test) to examine interannual differences in these indices. Furthermore, the Spearman correlation coefficient was used to evaluate the relationship between the diversity index and species abundance.

## 3. Results

### 3.1. Fish Species List

A total of 166 fish species belonging to 11 orders, 33 families, and 69 genera were documented in Meiji Reef from 1998 to 2018 (Table 2 and Appendix A). Among these, the dominant group comprised 128 Perciformes species from 20 families, accounting for a significant proportion of the cataloged species at approximately 77.10%. Additionally, there were also notable occurrences of Beryciformes with 11 species, Tetraodontiformes with 9 species, Anguilliformes with 5 species, and Beloniformes with 4 species. The remaining orders exhibited fewer than four recorded fish species.

At the family level, Serranidae exhibited the highest species richness with 19 species, followed by Labridae, Lethrinidae, Lutjanidae, Scaridae, Holocentridae, Mullidae, and Scolopsidae with 15 species, 14 species, 13 species, 13 species, 11 species, 10 species, and 8 species. Acanthuridae, Apodonidae, and Balistidae exhibited a moderate diversity with six fish species in each family. Muraenidiae, Carangideae, and Scombrideae were represented by four fish species each. Belonidiae, Chaetodontidiae, Pentapodidiae, and Siganidiae contributed three species each to the overall diversity. The remaining fifteen families had less than three recorded species.

According to the dependence of fish on coral reefs and their habitat preferences, there were 155 reef-dependent fish, accounting for 93.37% of the total. Additionally, there existed 11 reef-independent or pelagic species, constituting 6.63%. These included *Tylosurus acus*, *Tylosurus crocodilus*, *Tylosurus giganteus*, *Exocoetus volitans*, *Ruvettus pretiosus*, *Euthynnus affinis*, *Gymnosarda unicolor*, *Rastrelliger kanagurta*, *Scomber australasicus*, *Fistularia petimba*, and *Atherinomorus lacunosus* which inhabit the open ocean or upper ocean shelf.

The interannual variation in the species richness of fish on Meiji Reef observed from 1998 to 2018 is presented in Figure 2. The interannual variation in species richness was substantial. The dominant fish in the lagoon of Meiji Reef in the South China Sea during 1998–2018 is presented in Table 3. From 1998 to 2018, the dominant species of fish in Meiji Reef changed significantly, the number of dominant species decreased, and some parrotfish did not appear again. The number of Holocentridae and Mullidae species increased.

### 3.2. Inclusion Index at Taxonomic Level of Fish

The TINCL values of fish and their relationship with species number in Meiji Reef from 1998 to 2018 are depicted in Figure 3 and Table 4. Notably, the TINCL values for order, family, and genus were consistently high. Over the study period, the average TINCL values for (family, genus, species), (genus, species), and (species) within orders, families, and genera were recorded as (3.00, 6.27, 15.09), (2.09, 5.03), and (2.41), respectively. Furthermore, a positive correlation was observed between TINCL value and species number across different periods; indicating an increasing trend as species numbers rose. Significant correlations were found between order-level indices such as (genus, species) and family-level indices like (genus) with respect to species number according to Pearson correlation analysis at a significance level of *p* < 0.01. At the genus level, a comparison between years revealed that while the TINCL of fish remained similar between 1998–1999 and 2016–2018, other TINCL values during the former period were significantly lower than those observed during the latter period at a significance level of *p* < 0.05 (*T*-test).

### 3.3. Genus–Family Index

The family, genus, and G–F indices in Meiji Reef for different periods are given in Table 5. The F index, G index, and G–F index of fish from 1998 to 2018 were recorded as 15.15, 3.84, and 0.75, respectively. The G index exhibited the least variation across different periods, while the G–F index demonstrated the highest degree of variability. Significant variations were observed in the G–F index of fish between 1998–1999 and 2016–2018 in Meiji Reef. The F index, G index, and G–F index of fish exhibited a positive correlation with species richness (Figure 4). The variation in the G–F index was found to be influenced by differences in fish species richness. Both the F index and G index exhibited significant correlations with species richness (Pearson correlation, *p* < 0.05), while no significant correlation was observed between the standardized G–F index and species number (Pearson correlation, *p* > 0.05). The fish G index, F index, and G–F index in 1998 were the lowest among all periods, as was the species richness. However, the G–F index and F index of fish with more species in 2016–2018 were significantly higher than those with fewer species in 1998–1999 (*T*-test, *p* < 0.05). Interestingly, the G index remained similar to that of 1998–1999, indicating a relatively higher inter-genus diversity of fish during that period (*T*-test, *p* < 0.05).

### 3.4. Average Taxonomic Distinctness Index and Variation in Taxonomic Distinctness

The taxonomic distinctness indices of fish in Meiji Reef at different points between 1998 and 2018 are depicted in Figure 5, Figure 6 and Figure 7. The Δ^+^ and Λ^+^ values of fish from 1998 to 2018 were recorded as 56.1 and 148.5, respectively. During the periods of 1998–1999 and 2016–2018, the Δ^+^ and Λ^+^ values stood at 54.7 and 155.2, respectively, with a subsequent increase to 56.3 and 161.6 for the latter period. This upward trend indicated an augmented genetic disparity among fish species in Meiji Reef over time, accompanied by a reduction in species classification uniformity.

## 4. Discussion

### 4.1. Species Composition

According to our survey conducted from 1998 to 2018, the total number of fish species recorded on Meiji Reef amounted to 166, surpassing Zhubi Reef in the South China Sea by a margin of 14 species [40]. However, it fell short by 9 species compared to Qilianyu Island (mainly known as Zhaoshu Island) and lagged behind Taiping Island in the South China Sea by a significant difference of 300 species [41,42]. The fish richness on Meiji Reef exhibited similarities with Zhubi Reef and Qilianyu Island but remained considerably smaller than that found on Taiping Island. This disparity can be attributed primarily to variations in area size and habitat complexity among different islands and reefs, as well as disparities in the frequency of fish surveys conducted across these locations [43,44,45,46]. The diversity of coral reef fish is positively correlated with both the area and habitat complexity of a coral reef, particularly in larger areas [44].

Meiji Reef is a semi-closed atoll, allowing for water exchange between the lagoon and its surroundings. In contrast, Zhubi Reef is a closed atoll where water exchange occurs only during high tides due to the barrier of the reef flat. Qilianyu Island consists of seven small open islands and reefs, including Zhaoshu Island, which has been surveyed relatively infrequently. Taiping Island stands as the largest island in the South China Sea. Consequently, while Meiji Reef exhibits better fish habitat conditions compared to Zhubi Reef, it falls short when compared to Qilianyu Island and particularly Taiping Island. Apart from differences in species richness, variations could also be observed among coral reef fish communities regarding their composition of ecologically significant or indicative species. For instance, Meiji Reef hosted 13 parrotfish species along with 3 pomacentridae and 3 chaetodontidae species, whereas Zhubi Reef and Qilianyu Island had 5 and 11 parrotfish species, 16 and 21 pomacentridae species, and 8 and 14 chaetodontidae species, respectively [40,41]. These distinctions in characteristic coral reef fish served as important indicators reflecting ecological functional disparities among different coral reef fish. Taiping Island has a larger expanse and a more intricate habitat structure in comparison to Meiji Reef [42]. Consequently, it is imperative to enhance the preservation of effective coral reef habitats for the safeguarding of coral reef fish.

The interannual variation in species richness was substantial; however, it failed to accurately depict the true extent of species diversity due to the inherent differences in fishing efforts across years. The relative changes in species richness (species richness per unit of fishing effort) could serve as a robust indicator for comparing the true abundance of species over time [31,47]. Our research findings further demonstrated a discernible declining trend in relative species richness specifically observed on Meiji Reef. This could be attributed to the potential influence of various external factors, such as environmental changes and anthropogenic pressures, which could progressively alter the species composition of fish communities over time by reducing their redundancy [48,49,50]. Furthermore, it was noteworthy that reefs in the South China Sea, including Meiji Reef, have undergone substantial habitat degradation throughout the years, which might lead to changes in the taxonomic diversity index [24,31,51].

### 4.2. Inclusion Index at Taxonomic Level and Genus–Family Index

Compared to the Dongsha Islands, which had similar habitats, Meiji Reef exhibited a significantly lower diversity of fish species, which could be attributed to the greater extent of coral reef area present on the Dongsha Islands [52]. While the TINCL values at the genus level were comparable between the two regions, Meiji Reef demonstrated considerably reduced TINCL values at higher taxonomic levels such as orders and families [52]. Compared to bays with non-coral reef habitats, such as Sanmen Bay, Hangzhou Bay, Xiangshan Bay, Taizhou Bay, and Yueqing Bay in the East China Sea, Meiji Reef exhibited significantly higher fish species richness and TINCL values for each taxonomic order [53]. Compared with Daya Bay in the South China Sea, the number of fish species in Meiji Reef was much smaller than that in Daya Bay, but the TINCL values of its families and genera were higher than those in Daya Bay [22]. The TINCL of fish in Meiji Reef exhibited a consistently higher level compared to non-coral reef ecosystems, indicating that coral reefs, as the most biodiverse marine ecosystems, exhibit a greater level of fish diversity compared to other marine habitats. In the same season from 2017 to 2018, there was a significant decrease in the genera-level TINCL of fish in Meiji Reef when compared with 1998–1999. Additionally, the distribution at the genera level appeared more dispersed, and distant relatives were observed, potentially indicating a declining trend in species richness within Meiji Reef. At the genus level comparison between years revealed that while the TINCL of fish remained similar between 1998–1999 and 2016–2018; other TINCL values during the former period were significantly lower than those observed during the latter period. This discrepancy might be attributed to variations in fish species richness across these periods [31,47]. This was also supported by our analysis of the variation trend of TINCL indices with species richness on Meiji Reef in Figure 2. Interestingly, in May 2017, the genus-level TINCL was found to be lower compared to May 1999 despite having an equal number of fish species. The decline in coral cover might constitute a significant factor contributing to the reduction in genera-level TINCL. The live coral cover on the Meiji reef exhibited considerably lower levels during 2016–2018 compared to 1998–1999, which coincided with a concurrent decrease in fish species richness within reef-dependent genera such as parrotfish and butterfly fish [24,51].

The G–F index was widely employed in the study of avian and animal diversity, enabling a comparison of biodiversity at the family and genus levels across different regions [15,54,55]. In recent years, its applications have expanded to encompass marine biodiversity such as benthos, macroalgae, and fish [53,56,57]. Our research demonstrates a positive correlation between the number of fish species in Meiji Reef and corresponding increases in the G index, F index, and G–F index. This finding aligns with previous studies on taxonomic diversity among coral reef fish in the Xisha Islands by Li et al. [41,58], indicating that fish species richness played a crucial role in shaping the diversity of coral reef fish families and genera. Moreover, our analysis revealed significantly higher overall F-index values along with elevated G-index and G–F-index values for fish in Meiji Reef from 1998 to 2018 compared to those observed in Sanmen Bay as well as five other bays off Zhejiang [53], surpassing even the levels recorded for the Qilianyu area of the South China Sea during the 1970s [41]. These results were consistent with changes observed through TINCL analysis for fish within both areas, collectively suggesting that Meiji Reef exhibited relatively high family- and genus-level diversity among reef fish. These changes were indicative of alterations in the diversity index, such as the decline in parrotfish species, which could be attributed to anthropogenic activities like fishing and construction, as well as coral reef bleaching [24,51].

During 1998–1999, the F index closely approximated the G index; however, in 2017–2018, the F index exhibited a significantly higher value than the G index, indicating a relative increase in fish family diversity at Meiji Reef. The diversity of genera was relatively reduced, with species being predominantly limited to fewer families and even single families. The increase in fish family diversity and a decrease in genus diversity were observed during 1998–1999 and 2017–2018. This could be attributed to the higher concentration of species within fewer families or the presence of numerous monotypic families (Appendix A). For instance, on average, each family might have contained two genera and each genus might have had two species in 1998–1999; however, by 2017–2018, it was possible that each family contained only one genus, and each genus comprised just one species [15]. The standardized G–F index for fish at Meiji Reef during 1998–1999 was considerably lower compared to that of 2016–2018, suggesting a relative decrease in intergeneric diversity in 2017–2018 which aligned with the trend observed for the TINCL values of fish species at Meiji Reef. TINCL and G–F indices are commonly employed to assess taxonomic diversity across different regions based on presence or absence data. This study aims to compare fish diversity within the same area (Meiji Reef) across different time periods using these two indices. Given that both TINCL and G–F indices increased alongside fish species richness in each period, with some indices exhibiting significant positive correlations with species richness, it is important to consider how variations in fish species richness between different time periods may influence the application of these two diversity indices. Exploring methods to account for this influence would enhance comparative studies across historical periods. In the survey, increasing the sampling rate enabled the maintenance of fish species richness at a certain level. For instance, as species richness increased, there was no alteration in the G–F index, thereby facilitating a more accurate assessment of fish diversity through the G–F index.

### 4.3. Average Taxonomic Distinctness Index and Variation in Taxonomic Distinctness

The Δ^+^ of fish in Meiji Reef was slightly higher than that in Qilianyu (54.2) and slightly lower than that in Zhubi Reef (58.8), indicating that fish on Meiji Reef were slightly more closely related to each other than those on Qilianyu and slightly closer to each other than those on Zhubi Reef. However, the Λ^+^ of fish in Meiji Reef was much higher than that in Qilianyu Island and Zhubi Reef, indicating that the uniformity of the taxonomic relationship among fish in Meiji Reef was much lower than that in Qilianyu Island and Zhubi Reefs [23,40,41]. Therefore, the distribution of different coral reef fish communities in the South China Sea was uneven at each taxonomic level. This may be related to the geographical location of different reefs, coral reef topographic structure, effective habitat area, and complexity [43,44,59]. The distribution of coral reef fish species worldwide exhibits a latitudinal and longitudinal gradient, with its epicenter located within the Coral Triangle (encompassing reefs between Indonesia, the Philippines, Papua New Guinea, and the Solomon Islands) [44]. The structural complexity of coral reefs positively correlates with their provision of microhabitats, thereby promoting higher fish species richness [59]. Pinheiro et al. demonstrated that biogeographic factors exerted significant depth-dependent effects on coral reef fish species richness; specifically, deep coral reefs exhibited a decline in fish richness as depth increased [60]. For example, in terms of families containing more than 10 species of fish, Meiji Reef consisted of seven families, including serranidae (19 species), Labridae (15 species), Lethrinidae (14 species), Lutjanidae (13 species), Scaridae (13 species), Holocentridae (11 species), and Mullidae (10 species). There were seven families in Qilianyu Island, including Labridae (23 species), Pomacentridae (21 species), Holocentridae (18 species), Acanthuridae (14 species), Chaetodontidae (14 species), Scaridae (11 species), and Lutjanidae (10 species). Zhubi Reef consisted of 16 species of Pomacentridae, 15 species of Scaridae, and 10 species of serranidae. The differences between the three islands and reefs were obvious. Compared to other coral reef habitats, the Δ^+^ of fish in Meiji Reef was slightly higher than that of San Jose Island (55.1) [61]. According to Shi et al. [52], the lower the latitude on a large scale, the smaller the Δ^+^ of marine fish in China, and the closer the relationship between fish species. The study of Li et al. [21] showed that high species diversity and low taxonomic diversity were the main characteristics of large-scale distribution patterns of coral reef fish diversity in the South China Sea.

Graham et al. [62] demonstrated a declining trend in the Δ^+^ and Λ_+_ of reef fish across various substrate reef habitats in the inner islands of Seychelles following coral bleaching. However, when compared to 1998–1999, Δ^+^ and Λ^+^ exhibited an increase among reef fish species in Meiji Reef during 2016–2018, indicating a greater dissimilarity between fish species and reduced taxonomic uniformity. This observation may be attributed to differences in fish composition and taxonomic order between the inner islands of the Seychelles and Meiji Reef. Analysis of the G–F index for fish in Meiji Reef during both periods revealed that fish species were limited to fewer families, with several monotypic families present as well. Inter-genus diversity was significantly diminished, resulting in a restricted range of genera among the fish species inhabiting Meiji Reef. These findings provide substantial evidence supporting increased dissimilarity among fish species at Meiji Reef along with decreased taxonomic uniformity.

In recent decades, coral reefs have experienced severe degradation and are at risk of decline due to environmental changes and the increasing impact of human activities [63]. This degradation has resulted in a decrease in fish diversity and local extinction, further weakening the stability and recovery potential of coral reefs [45]. The maintenance of coral reef fish diversity and their ecological functions is crucial for enhancing the resistance and resilience of these ecosystems against threats, which is a global concern [64]. Our research findings indicated that from 1998 to 2018, there was a reduction in taxonomic diversity among fish species in Meiji Reef, an increase in species dissimilarity, and a decrease in taxonomic consistency. These trends aligned with the global pattern observed for coral reef fish diversity. The decline in reef fish diversity has been accompanied by reduced coral cover and habitat degradation. Therefore, it is imperative to strengthen the protection measures for coral reef fish as this would contribute toward ecosystem restoration.

## 5. Conclusions

The species richness of fish varied among different coral reefs in the South China Sea. This variation was due to differences in reef area and habitat complexity, as well as potential variations in survey frequency across reefs. Therefore, further investigations are needed to comprehensively compare fish diversity among different coral reefs. In Meiji Reef, taxonomic index values for families, genera, and species of fish were higher. Meiji Reef generally exhibited higher TINCL values compared to non-reef ecosystems. However, comparing data from 1998–1999 with that from 2017–2018 revealed a significant decrease in genera TINCL for fish at Meiji Reef during the same season, indicating a declining trend in species richness. The fish species richness played an important role in shaping the diversity of coral reef fish families and genera. In contrast to 1998–1999, the diversity of fish families increased relatively at Meiji Reef during 2017–2018; however, the diversity of genera decreased relatively, and many species were limited to fewer or even single families. Distribution patterns among different coral reef fish communities in the South China Sea exhibited unevenness across taxonomic levels. The comparison of data between 1998–1999 and 2016–2018 showed an increase in Δ^+^ and Λ^+^ indices within reef fish in Meiji Reef, indicating greater dissimilarity among species relationships and reduced taxonomic uniformity. These trends aligned with the global pattern observed for coral reef fish diversity. The decline in reef fish diversity has been accompanied by reduced coral cover and habitat degradation. Therefore, it is imperative to strengthen the protection measures for coral reef fish as this would contribute toward ecosystem restoration. Implementing fishing bans, enhancing environmental quality, establishing protected areas, and other similar actions can be considered.

## Figures and Tables

**Figure 1 biology-13-00740-f001:**
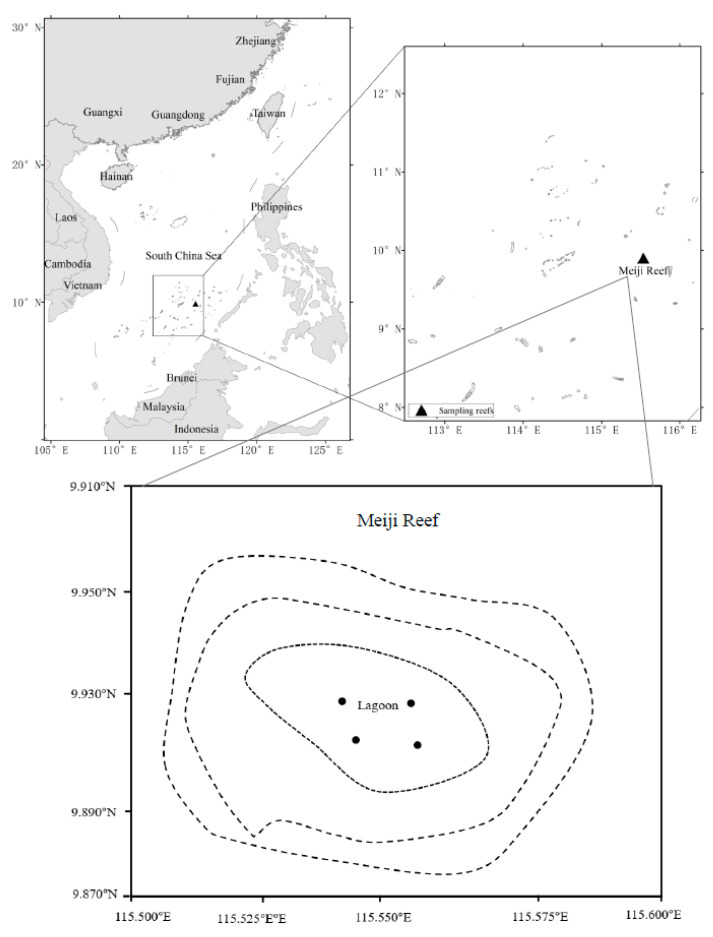
Study area (Meiji Reef lagoon) in the South China Sea. ●: approximate fishing sites.

**Figure 2 biology-13-00740-f002:**
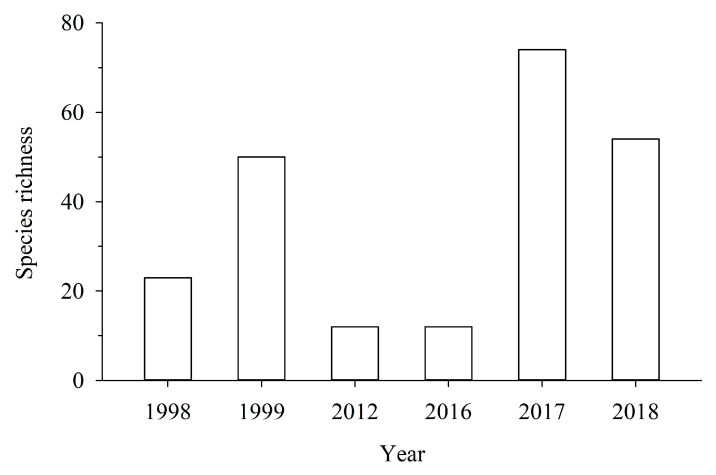
Interannual variation in species richness of reef fish on Meiji Reef observed from 1998 to 2018.

**Figure 3 biology-13-00740-f003:**
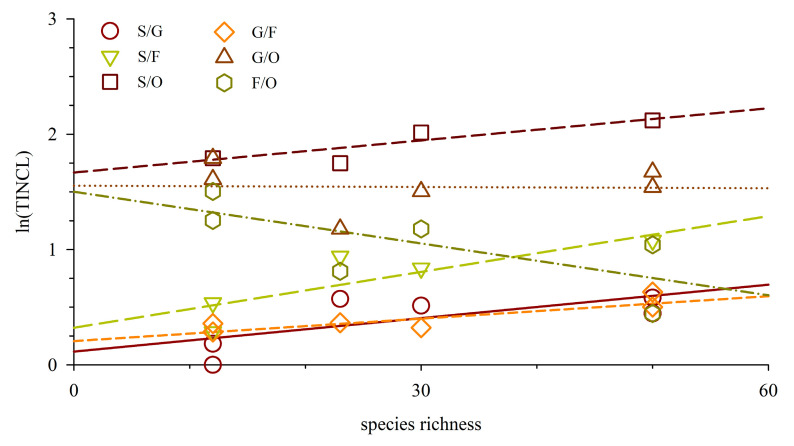
Variation trend of TINCL indices with species richness on Meiji Reef. S, G, F, and O denote species, genus, family, and order.

**Figure 4 biology-13-00740-f004:**
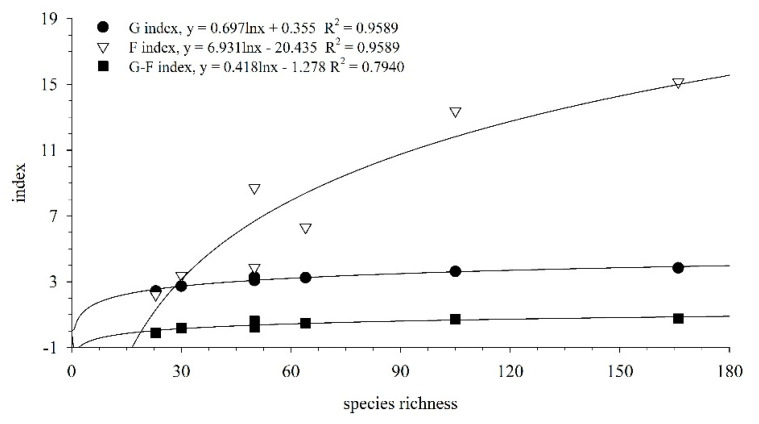
G index, F index, and G–F index increasing with species richness on Meiji Reef.

**Figure 5 biology-13-00740-f005:**
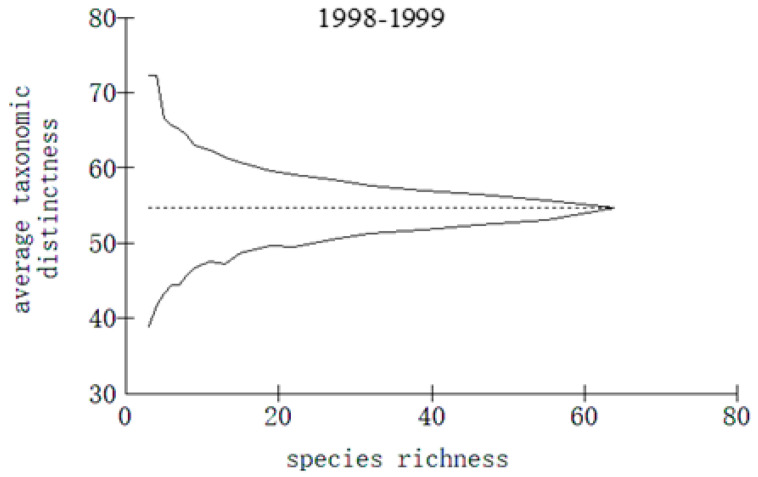
Average taxonomic distinctness and variation in taxonomic distinctness of reef fish on Meiji Reef in 1998–1999.

**Figure 6 biology-13-00740-f006:**
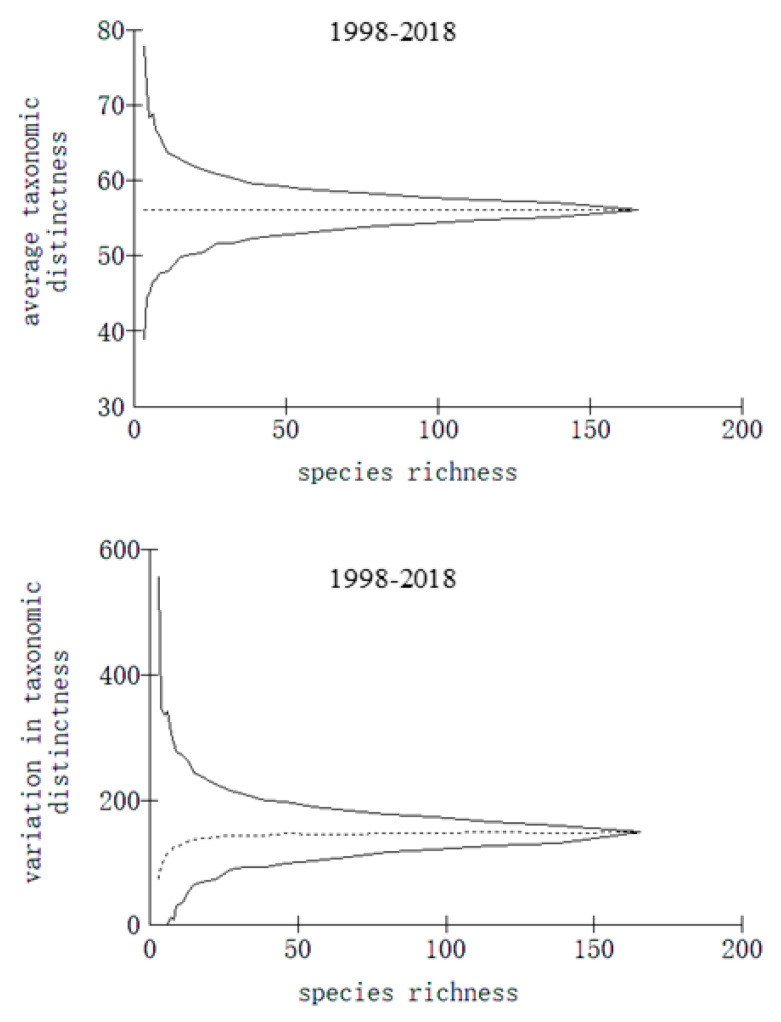
Average taxonomic distinctness and variation in taxonomic distinctness of reef fish on Meiji Reef in 1998–2018.

**Figure 7 biology-13-00740-f007:**
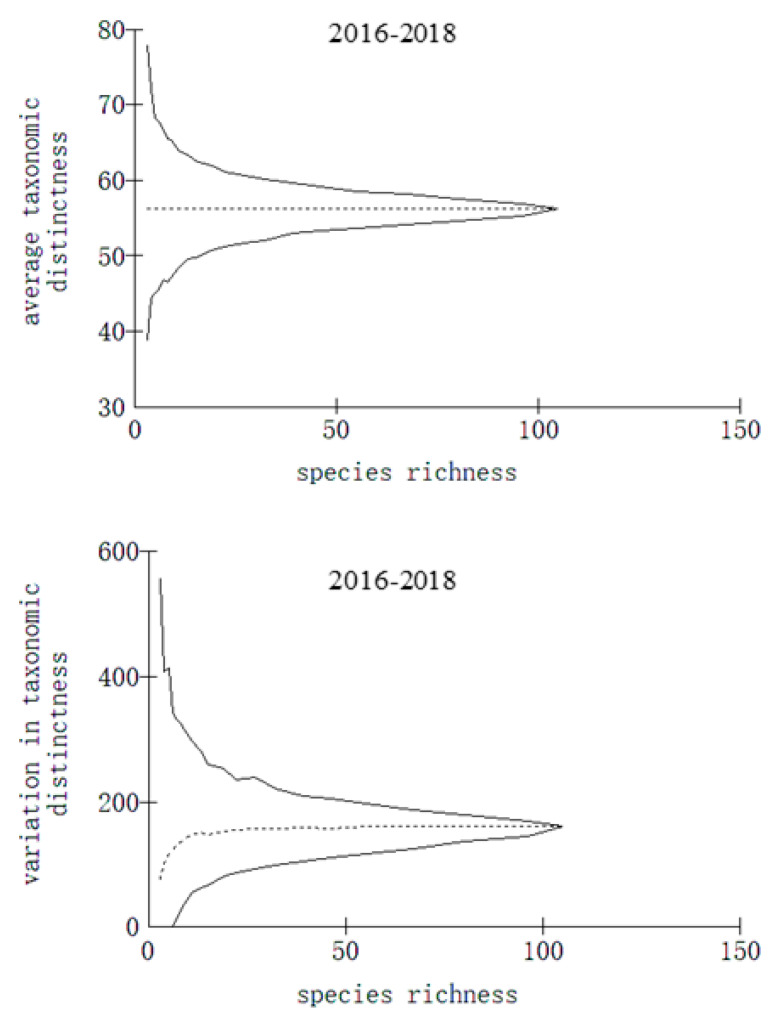
Average taxonomic distinctness and variation in taxonomic distinctness of reef fish on Meiji Reef in 2016–2018.

**Table 1 biology-13-00740-t001:** Sampling information of reef fish for Meiji Reef, South China Sea.

Year–Month	Sampling Region	Net and Specifications
1998–05	reef rim, lagoon	hand line, hook with barb, size: 26.0 mm total length, 10.0 mm gape size, 1.0 mm thickness, 0.33 mm diameter nylon wire; longline, main line: ϕ10 mm floating Lux wire, float: ϕ10 mm PVC, branch line: 150 px 3 Lux wire, 11 m long, 16 m spacing, hook: ϕ4 mm stainless steel, 60 mm long, 30 mm wide
1999–05	reef rim, lagoon	hand line (same specifications as May 1998); gillnet: 40 m length, 1.1 m height, 5.5 cm mesh size, 0.20 mm diameter nylon wire; longline (same specifications as May 1998)
2012–09	reef flat, lagoon	gillnet: 50 m length, 1.5 m height, 3.3 cm inner mesh size, 7.8 cm inner mesh size, 0.20 mm diameter nylon wire
2016–05	reef rim, lagoon	gillnet: (same specifications as September 2012)
2017–05	reef rim, lagoon	hand line (same specifications as May 1998); gillnet: (same specifications as May 2016)
2017–12	reef rim, lagoon	hand line (same specifications as May 1998); gillnet: (same specifications as May 2016)
2018–05	reef rim, lagoon	hand line (same specifications as May 1998); gillnet: (same specifications as May 2016)
2018–09	reef rim, lagoon	hand line (same specifications as May 1998); gillnet: 50 m length, 1.1 m height, 2.0 cm inner mesh size, 7.0 cm inner mesh size, 0.20 mm diameter nylon wire

**Table 2 biology-13-00740-t002:** Compositions of order, family, and genera for reef fish on Meiji Reef, South China Sea.

Order	Species Richness	%	Family	Species Richness	%	Genus	Species Richness	%
Carcharhiniformes	1	0.60	Scyliorhinidae	1	0.60	*Cephaloscyllium*	1	0.60
Myliobatiformes	3	1.81	Dasyatidae	3	1.81	*Maculabatis*	1	0.60
						*Neotrygon*	1	0.60
						*Taeniurops*	1	0.60
Anguilliformes	5	3.01	Muraenesocidae	1	0.60	*Muraenesox*	1	0.60
			Muraenidae	4	2.41	*Gymnothorax*	4	2.41
Atheriniformes	1	0.60	Atherinidae	1	0.60	*Atherinomorus*	1	0.60
Aulopiformes	2	1.20	Synodontidae	2	1.20	*Synodus*	1	0.60
						*Saurida*	1	0.60
Beloniformes	4	2.41	Belonidae	3	1.81	*Tylosurus*	3	1.81
			Exocoetidae	1	0.60	*Exocoetus*	1	0.60
Beryciformes	11	6.63	Holocentridae	11	6.63	*Myripristis*	3	1.81
						*Neoniphon*	1	0.60
						*Sargocentron*	7	4.22
Gasterosteiformes	1	0.60	Fistulariidae	1	0.60	*Fistularia*	1	0.60
Perciformes	128	77.10	Acanthuridae	6	3.61	*Acanthurus*	3	1.81
						*Ctenochaetus*	1	0.60
						*Naso*	2	1.20
			Apogonidae	6	3.61	*Apogon*	4	2.41
						*Apogonichthyoides*	1	0.60
						*Cheilodipterus*	1	0.60
			Carangidae	4	2.41	*Carangoides*	2	1.20
						*Caranx*	2	1.20
			Chaetodontidae	3	1.81	*Chaetodon*	3	1.81
			Gempylidae	1	0.60	*Ruvettus*	1	0.60
			Gerreidae	1	0.60	*Gerres*	1	0.60
			Labridae	15	9.04	*Anampses*	3	1.81
						*Bodianus*	2	1.20
						*Cheilinus*	4	2.41
						*Choerodon*	1	0.60
						*Epibulus*	1	0.60
						*Halichoeres*	4	2.41
			Lethrinidae	14	8.43	*Lethrinus*	14	8.43
			Lutjanidae	13	7.83	*Lethrinus*	2	1.20
						*Aphareus*	8	4.82
						*Lutjanus*	1	0.60
						*Macolor*	1	0.60
						*Paracaesio*	1	0.60
			Mullidae	10	6.02	*Pristipomoides*	1	0.60
						*Mulloidichthys*	5	3.01
						*Parupeneus*	4	2.41
			Pentapodidae	3	1.81	*Upeneus*	1	0.60
						*Gnathodentex*	2	1.20
			Pinguipedidae	2	1.20	*Pentapodus*	2	1.20
			Pomacentridae	2	1.20	*Parapercis*	1	0.60
						*Amblyglyphidodon*	1	0.60
			Scaridae	13	7.83	*Teixeirichthys*	1	0.60
						*Calotomus*	1	0.60
						*Cetoscarus*	11	6.63
			Scolopsidae	8	4.82	*Scarus*	8	4.82
			Scombridae	4	2.41	*Scolopsis*	1	0.60
						*Euthynnus*	1	0.60
						*Gymnosarda*	1	0.60
						*Rastrelliger*	1	0.60
			Serranidae	19	11.45	*Scomber*	1	0.60
						*Anyperodon*	9	5.42
						*Cephalopholis*	7	4.22
						*Epinephelus*	2	1.20
			Siganidae	3	1.81	*Plectropomus*	3	1.81
			Sparidae	1	0.60	*Siganus*	1	0.60
Pleuronectiformes	1	0.60	Bothidae	1	0.60	*Monotaxis*	1	0.60
Tetraodontiformes	9	5.42	Tetraodontidae	1	0.60	*Bothus*	1	0.60
			Balistidae	6	3.61	*Arothron*	1	0.60
						*Balistapus*	1	0.60
						*Balistoides*	1	0.60
						*Melichthys*	1	0.60
						*Pseudobalistes*	2	1.20
			Diodontidae	2	1.20	*Sufflamen*	2	1.20

**Table 3 biology-13-00740-t003:** Dominant fish in the lagoon of Meiji Reef in the South China Sea during 1998–2018.

Sampling Gear	Year–Month	Dominant Species
Gillnet	1999–05	*Scarus prasiognathos*, *Scarus sordidus*, *Acanthurus nigrofuscus*, *Scarus longiceps*, *Scarus dimidiatus*, *Siganus fuscescens*
	2016–05	*Mulloidichthys vanicolensis*, *Lutjanus kasmira*, *Lutjanus erythropterus*
	2017–05	*Lutjanus gibbus*, *Parupeneus barberinus*
	2018–05	*Taeniurops meyeni*, *Diodon liturosus*, *Lutjanus kasmira*, *Mulloidichthys vanicolensis*
Hand line	1998–05	*Aphareus rutilans*, *Sargocentron spiniferum*, *Monotaxis grandoculis*, *Myripristis murdjan*, *Lethrinus kalloperus*, *Lethrinus lentjan*, *Cephalopholis aurantius*
	1999–05	*Cephalopholis urodelus*, *Cephalopholis pachycentron*, *Cetoscarus ocellatus*, *Apogon trimaculatus*, *Adioryx furcatus*, *Epinephelus merra*, *Sargocentron spiniferum*, *Siganus canaliculatus*, *Adioryx caudimaculatus*
	2017–05	*Lutjanus kasmira*, *Lutjanus sebae*, *Epinephelus merra*
	2018–05	*Epinephelus merra*, *Pentapodus caninus*, *Lethrinus atkinsoni*, *Rastrelliger kanagurta*, *Lethrinus rubrioperculatus*

**Table 4 biology-13-00740-t004:** Comparison of inclusion indices at taxonomic level on Meiji Reef for different periods.

Period	Species Richness	S/G	S/F	S/O	G/F	G/O	F/O
1998–2018	166	2.41	5.03	15.09	2.09	6.27	3.00
1998–1999	64	1.94	3.37	9.14	1.74	4.71	2.71
2016–2018	105	1.98	4.04	13.13	2.04	6.63	3.25
1998–05	23	1.77	2.56	5.75	1.44	3.25	2.25
1999–05	50	1.79	2.94	8.33	1.65	4.67	2.83
2012–09	12	1.00	1.33	6.00	1.33	6.00	4.50
2016–05	12	1.20	1.71	6.00	1.43	5.00	3.50
2017–05	50	1.56	2.94	8.33	1.88	5.33	1.56
2018–05	30	1.67	2.31	7.50	1.38	4.50	3.25

Note: S, G, F, and O represent species, genus, family, and order, respectively.

**Table 5 biology-13-00740-t005:** G–F indices of reef fish on Meiji Reef for different periods.

Period	Species Richness	G Index	F Index	G–F Index
1998–2018	166	3.84	15.15	0.75
1998–1999	64	3.24	6.30	0.49
2016–2018	105	3.62	13.39	0.73
1998–05	23	2.43	2.21	−0.10
1999–05	50	3.07	3.86	0.21
2012–09	12	2.254	3.466	0.350
2016–05	12	2.345	2.485	0.156
2017–05	50	3.28	8.71	0.62
2018–05	30	2.73	3.38	0.19

## Data Availability

The original contributions presented in the study are included in the article. The data presented in this study are available on request from the corresponding author.

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
