# Peer review of "Taxonomic Diversity and Interannual Variation of Fish in the Lagoon of Meiji Reef (Mischief Reef), South China Sea"

_biology, 2024, doi:10.3390/biology13090740_

Round 1
Reviewer 1 Report
Comments and Suggestions for Authors
Review for the paper "Taxonomic diversity and interannual variation of fishes in Meiji Reef, South China Sea" by Yuyan Gong, Jun Zhang, Zuozhi Chen, Yancong Cai, Yutao Yang submitted to “Biology”.
The authors of this research paper conducted an analysis of reef-dwelling fish species at Meiji Reef, a significant tropical semi-enclosed atoll in the South China Sea known for its rich marine biodiversity. This study used data gathered from various fishing methods, including hand-fishing, line-fishing, and gillnet surveys, over a span of twenty years, from 1998 to 2018. The focus was on summarizing the species composition and diversity of fish in this region, highlighting important taxonomic indices that provide insights into the ecological dynamics of the reef. They found that a total of 166 reef-dwelling fish species were recorded at Meiji Reef during the study period. These species belonged to 69 genera, 33 families, and 11 orders, with the majority classified under the order Perciformes, representing 77% of the total species cataloged. The researchers distinguished between typical resident species, which accounted for 93%, and atypical wandering species, at just 7%. This classification underscores the reliance of most fish species on coral reef habitats for survival, emphasizing the ecological importance of these areas.
The authors noted that various diversity indices, such as the taxonomic inclusion index and the genus-family diversity index, indicated high levels of diversity within the fish populations of Meiji Reef. Interestingly, the average taxonomic distinctness index of fish was found to be higher in the earlier years of the study (1998-1999) than in the later years (2016-2018), suggesting potential shifts in species composition or abundance over time.
This research enhances our understanding of the complex dynamics governing the fish species in Meiji Reef and the broader implications for biodiversity conservation in tropical marine ecosystems. It highlights the necessity for ongoing monitoring and assessment of reef fish populations to promote effective conservation strategies, especially in the face of changing environmental conditions and human impacts.
Despite the importance of the topic, the paper lacks important information on statistical analysis. In addition, the paper is currently primarily descriptive in nature, lacking an adequate discussion of the authors' data, which limits its overall appeal to an international readership. In order to meet the criteria for exceptional papers, the authors are advised to substantially revise and update their work.
Recommendations.
Abstract.
L 30. The authors mention "average taxonomic distinctness index" for both Δ+ and Λ+. Please correct. The latter should read "variation in taxonomic distinctness".
In the abstract, the authors should indicate the significance of their results for monitoring and conservation.
Introduction.
L 47. The authors should provide more detailed background information on coral reefs. An explanation of the ecological roles of coral reefs and the specific functions of reef fishes within these systems would provide a better context for why their diversity is critical.
L 51-52. The authors should provide examples of how declines in fish diversity affect the ecosystem services provided by coral reefs.
L 75. The authors should clarify how taxonomic diversity complements traditional species diversity indices in practical applications and provide examples or case studies of how the application of taxonomic diversity indices, such as the TINCL or G-F index, has improved assessments of coral reef fish communities compared to traditional indices.
L 79-82. The authors should report on key ecological characteristics of coral reefs in the South China Sea and explain how these relate to fish diversity, and include a brief overview of previous studies on coral reef fish diversity in the South China Sea, especially those focusing on taxonomic diversity.
Methods.
L 100. The authors should clarify the significance of the selected study years (e.g., 1998, 2012, 2016-2018) in terms of different environmental conditions or events (such as El Niño/La Niña periods) that may affect coral reef health.
L 101. The authors should clarify the rationale for using different types of fishing gear and how this might affect the diversity of species caught.
L 116. The authors should clarify the criteria that guided the selection of four sampling sites to provide insight into how representative the collection was of the overall fish community in Meiji Reef.
The authors should clarify the criteria used to determine the classification of fish species into typical resident and atypical reef-dwelling species, including their characteristics or behaviors that distinguish typical resident species from atypical species, with relevant citations.
L 214-218. The authors mention "correlations" in the results, but this methodology was not described in the materials and methods, nor was testing of the data for the necessary assumptions in the case of Pearson correlations.
L 218-221. What method was used to compare TINCL values between different time periods?
Results.
Section 3.1.
The authors should include information on inter-annual variation in species richness or composition observed from 1998 to 2018.
It would be useful to include illustrations of dominant species and families to enrich the text and aid in species identification.
Additional information on the conservation status or threats faced by any of the recorded species would highlight the importance of conservation efforts in Meiji Reef and surrounding areas. The authors should include this data in the Supplementary Material.
Section 3.2.
L 221. "This discrepancy may be attributed to variations in fish species abundance across these periods". This sentence corresponds to discussion and should be moved in that section. Also, the authors should provide more information of fish abundance in these two periods with relevant citations.
L 223-224. This sentence also corresponds to the discussion and needs detailed explanation. What might explain the discrepancy in genus-level TINCL values between May 1999 and May 2017, despite the same number of fish species? Were any external factors (e.g., environmental changes, anthropogenic pressures) considered that might have influenced the TINCL values over the years?
Discussion.
L 277. The authors should clarify how differences in area size and habitat complexity specifically affect fish biodiversity and community composition among these reefs.
L 278-291. It is important to discuss the potential impact of any human activities (such as fishing practices, pollution, or tourism) and/or environmental changes that might have impacted the fish populations and habitat conditions in Meiji Reef and other comparison sites.
In terms of conservation efforts and management strategies in the region, it would be useful to discuss the implications for Meiji Reef of having fewer species compared to Taiping Island.
L 293-294. The authors should explain the reasons for the significantly lower fish diversity at Meiji Reef compared to Dongsha Islands despite similar habitats.
L 302-302. The authors should explain why the TINCL of fishes in Meiji Reef exhibited a consistently higher level compared to non-coral reef ecosystems and how this higher value in reflect the ecological health of the reef.
L 303-305. What specific trends or factors were identified that led to the significant decrease in genera-level TINCL from 1998-1999 to 2017-2018?
L 308-322. What specific factors contributed to the observed variability in the G-F index between 1998-1999 and 2016-2018?
L 323-338. The authors should explain how the positive correlation between fish species number and the G-F index can inform future research and monitoring strategies.
What factors contributed to the relative increase in fish family diversity but a decrease in genera diversity observed at Meiji Reef between 1998-1999 and 2017-2018?
L 347-348. The authors should clarify how the geographical location and topographic structure of the reefs might influence the observed patterns of taxonomic dissimilarity and uniformity among reef fish communities.
In a broader context, authors should discuss how their results relate to global trends in marine biodiversity and the importance of maintaining fish diversity for ecosystem resilience and health.
Specific remarks
L 183. Consider replacing “Results and Analysis” with “Results”
L 266. Consider replacing “2016-2016” with “2016-2018”
Comments on the Quality of English LanguageMinor revision
Reviewer 2 Report
Comments and Suggestions for Authors This paper reports on the fish diversity of Meiji Reef in the South China Sea, covering a long survey period. The data is significant for the conservation of fish diversity in the region. However, the conservation status and endangered levels of the surveyed fish species are not provided in the article. It is recommended to include this information.Additionally, here are some suggested revisions for the entire manuscript:
Line 41-42: The abstract is missing a conclusion, including conservation recommendations and other related content.
Line 47-60: Since the introduction mentions the importance of traditional diversity indices for evaluating diversity, why were these traditional indices not used in this study to assess coral reef fish diversity? If they were not used, this aspect needs to be revised and corrected.
Line 91-98: Including a map of sampling points would help readers better understand the geographic area of the survey.
Line 109-120: Were species identified on-site, or were all collected specimens preserved and later identified in the laboratory? Did the sampling intensity potentially disrupt fish populations? Please provide a reasonable explanation in the text. Additionally, please include the permit details and number for the project in the Materials and Methods section.
Line 184: For different species, can you include photographs of each species or specimens in the supplementary materials to aid reader identification?
Line 186: Please list in Appendix A the species surveyed in 1988-1999, 2012, and 2017-2018.
Line 228: Could the decrease in species numbers be related to inconsistent survey intensity? Given the long time span, the survey teams might not have been the same. Could this factor have influenced the results? It is suggested to discuss this issue in the discussion section.
Line 375: The conclusion is missing conservation recommendations and suggestions.
Round 2
Reviewer 1 Report
Comments and Suggestions for Authors
The authors have provided sufficient responses to my comments.
Author Response
Comments: The authors have provided sufficient responses to my comments.
Response: Thank you again for reviewing this manuscript.
Reviewer 2 Report
Comments and Suggestions for Authors
The current version looks good overall; my only suggestion is to use a high-resolution image for Figure 1, as the current one is not clear enough.
Author Response
Comments: The current version looks good overall; my only suggestion is to use a high-resolution image for Figure 1, as the current one is not clear enough.
Response:Thank you for pointing out this. I have attached the sampling map in PDF format.
